# GAUSSIAN MASKED AUTOENCODERS

## ABSTRACT

This paper explores Masked Autoencoders (MAE) with Gaussian Splatting. While reconstructive self-supervised learning frameworks such as MAE operate on low-level pixels, the image synthesis community has evolved to use latent, *mid-level* representations for better generative visual data modeling. Our approach, named Gaussian Masked Autoencoder, or GMAE, aims to reconcile these two and get the benefits of both worlds. Like MAE, it reconstructs the image end-to-end in the pixel space; but beyond MAE, it also introduces an intermediate, 3D Gaussian-based representation and renders images via splatting. We show that GMAE can enable various zero-shot learning capabilities (*e.g.*, figure-ground segmentation, image layering, edge detection, *etc.*) while preserving the high self-supervised representation quality from MAE. To our knowledge, we are the *first to employ Gaussian primitives in an image representation learning framework* beyond optimization-based single-scene reconstructions. We believe GMAE will inspire further research in this direction and contribute to developing next-generation techniques for modeling high-fidelity visual data.

## 1 INTRODUCTION

Vision systems, by nature, process raw, low-level observations of the world, but visual reasoning frequently requires higher-level semantic abstractions of the data as well as spatial and geometric understanding. In this work we aim to effectively learn both semantic and geometric representations by learning masked auto encoding controlled by 3D Gaussians.

Learning high level semantics, can be achieved by supervised learning (Krizhevsky et al., 2012; He et al., 2015; Dosovitskiy et al., 2020) or by learning binding from large scale paired datasets (Radford et al., 2021; Jia et al., 2021; Zhang et al., 2022; Rombach et al., 2022). One promising approach in this direction is self-supervised learning Oquab et al. (2023); Grill et al. (2020); Bao et al. (2021); He et al. (2022); Wei et al. (2022). Initial efforts have tapped different tokens such as VAE (Ramesh et al., 2021; Kingma, 2013; Bao et al., 2021) or manually designed features Dalal & Triggs (2005); Wei et al. (2022). Masked Autoencoders (MAE) (He et al., 2022) demonstrated that representation learning could be just as effective by directly predicting the RGB values of masked patches. By directly predicting pixels, MAE enjoys both *simplicity* – no complication in staged training or dependence on external models, and *end-to-end optimizability* – a fundamental idea that underpins the success of deep learning.

On the other hand, visual reasoning also needs spatial understanding, and 3D awareness. To address the spatial and 3D understanding required for visual reasoning, extensive research has explored learning 3D representations from both single-view and multi-view setups (Kar et al., 2015; Tulsiani et al., 2017; Yariv et al., 2020; Yao et al., 2018; Riegler & Koltun, 2020). Methods based on single-view learning leverage prior knowledge to reconstruct 3D geometry from limited input, often relying on implicit representations (Tatarchenko et al., 2019; Yu et al., 2021; Li et al., 2020). Multi-view approaches, such as those utilizing photometric consistency, have demonstrated strong performance in 3D reconstruction by fusing spatial cues from multiple perspectives (Mildenhall et al., 2020; Kerbl et al., 2023). These advances form the foundation for integrating spatial understanding.

In this paper, we propose to learn high-level semantics and to have geometric understanding jointly via self-supervised learning. Our idea is conceptually simple: we train a masked auto-encoder pretraining, which has shown promising results on learning high level semantic tasks, along with 3D Gaussians (Kerbl et al., 2023) as intermediate representations to learn spatial and geometric understanding.

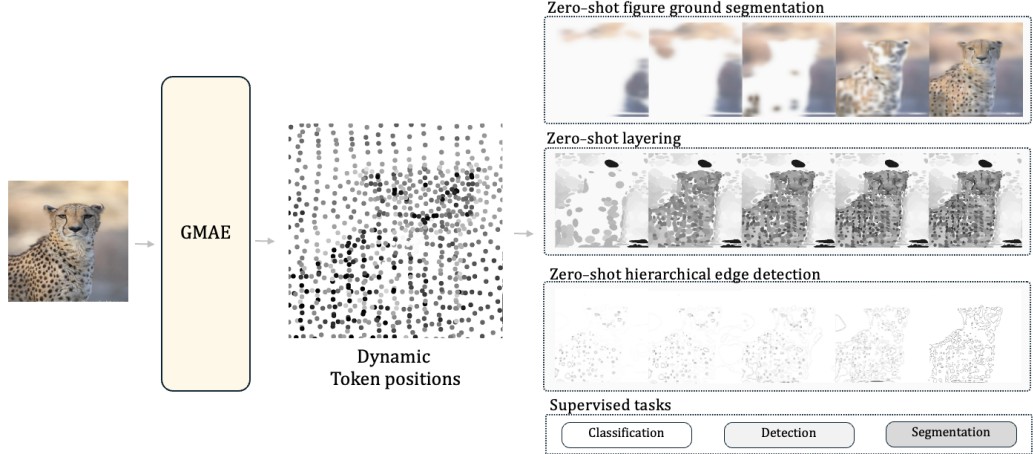

Figure 1: **Gaussian Masked Autoencoders (GMAE)** maintains high performance in supervised representation learning tasks such as classification, detection, and segmentation, but more importantly enables zero-shot capabilities. GMAE introduces a learned mid-level intermediate representation of 3D Gaussians that we train using pixel-based image reconstruction losses rather than direct supervision by rendering the Gaussians into pixel space. Through this reconstruction loss, the Gaussian collection learns to distribute non-uniformly in space and scale, dynamically following the input image's information density and high-frequency details. Having the degree of freedom in depth allows the model to learn the layering of objects and scenes, which enables figure-ground separation, layering, and edge detection without any training.

3D Gaussians were initially proposed for optimization-based 3D reconstruction. Different from geometrically uniform representations like square pixel patches, their size, location, and information distribution over the image are dynamically learned. Moreover, Gaussian-based representations could lend themselves to end-to-end learning thanks to splatting image rendering (Kerbl et al., 2023) that maps them back to the pixel space. We can, therefore, jointly learn such a mid-level representation within self-supervised frameworks such as MAE. We name our approach Gaussian Masked Autoencoders, or GMAE. To the best of our knowledge, *we are the first to explore such Gaussian primitives in a visual representation learning framework*, rather than an optimization-based 3D reconstruction framework for single scenes. Our approach adds only a negligible overhead compared to standard MAE training – the addition of splatting increases compute time by 1.5%. Without compromising representation-learning performance, GMAE gains significant wins in zero-shot capabilities.

Fig. 1 shows several built-in advantages of a Gaussian-based image representation. First, we note that the non-uniformity of the representation leads to a spatial distribution of representational density that correlates with the information density in the image. By allowing the 3D Gaussians to move along the $z$-axis, our model learns underlying structure of the natural world by observing not only the single viewpoint of one image but millions of such single views. As a result, we can find figure-ground segmentation, simple layering, and edge detection as depth discontinuity without any tuning.

Besides these advantages, we show that the representations learned with GMAE perform similarly to MAE on image classification and object detection tasks. The representation quality improves with the number of Gaussians used. These results suggest GMAE *augments* MAE and can serve as a better alternative in applications that can benefit from using mid-level representations. The advantage of GMAE becomes apparent when considering that splatting-based rendering is highly efficient, and our current training is *almost as fast as* vanilla MAE baselines.

We hope our exploration can inspire further research along this hybrid direction for representation learning: the reconstruction target is grounded to pixels while jointly learning effective mid-level representations for images. We believe it can contribute to the next generation of techniques for modeling high-fidelity visual data.

## 2 RELATED WORK

**Self-supervised Learning:** Over the years, self-supervised pre-training has proven effective in many areas, including language, vision, and robotics. In computer vision, there are two primary schools of thought: discriminative and reconstructive pre-training with visual data. Discriminative pre-training involves training an instance discrimination model to learn the similarity between different augmented image versions. Wu et al. (2018) and SimCLR (Chen et al., 2020) showed that instance discrimination training can be used to learn strong discriminative features via contrastive learning. Later, notable works such as MoCo (He et al., 2020) and DINO (Caron et al., 2021) have shown the effectiveness of such trained visual representations for various downstream tasks.

Reconstructive pre-training learns to model the data distribution by trying to reconstruct an image or a video from its noisy version. One of the most successful methods for such pre-training in computer vision has been the BERT (Devlin et al., 2018)-style masked modeling of images proposed by BEiT (Bao et al., 2021), and MAE (He et al., 2022). Compared to BERT, MAE uses asymmetric encoder-decoders, allowing it to be very efficient at training with high masking ratios. This style of reconstructive pre-training learns strong visual priors and shows impressive results on various downstream tasks such as object detection (Li et al., 2022), pose estimation (Xu et al., 2022), and robot tasks (Radosavovic et al., 2022).

**Mid-level Representations:** Image can be constructed by operating functions on some representations. One line of approachs keep the representations in the latent spaces, and use a pretrained decoder network to re-construct the image. VAE (Kingma, 2013) with image synthesis (Rombach et al., 2022; Li et al., 2023) are good examples of this case, along with MAE He et al. (2022) and BEiT Bao et al. (2021). Other line of approaches follow a structured representations to represent an image. There are various such options: super-pixels, Gaussians, SVG code and multi-plane images etc. For example Super-pixel sampling networks (Jampani et al., 2018) learns to predict super-pixels as the representation to reconstruct and to predict segmentations and flow. Multi-plane images is another way to represent an image (Tucker & Snavely, 2020), where an image is composed by multiple layered planes, and can be learned end-to-end. There are hybrid approaches also exist. For example, slot-attention (Locatello et al., 2020), learns an intermediate representation for objects, by adding a bottleneck in the model architecture. Similarly Leopart (Ziegler & Asano, 2022) learns to cluster the patches based on self-supervised clustering. In this paper, we take another approach which uses 3D Gaussians as intermediate representations to reconstruct an image.

**Gaussian Splatting:** Gaussian splatting (Kerbl et al., 2023) is a novel differentiable volume splatting technique using Gaussian primitives as the 3D representation, offering high optimization flexibility and high fidelity in reconstruction. This idea follows a long list of differentiable rendering techniques that recently gained significant attention as a method to bridge the gap between the 3D world and 2D images. Differentiable rendering allows for reconstructing 3D representations (e.g., meshes, point clouds) from 2D image signals by enabling gradient-based optimization. For example, (Liu et al., 2019) introduces a differentiable rasterizer for meshes using probability aggregation. (Lassner & Zollhöfer, 2021) proposes an efficient and differentiable formula to render large sets of point clouds. (Mildenhall et al., 2021) and (Barron et al., 2022) apply differentiable volume rendering (Levoy, 1990) to reconstruct 3D radiance fields from a handful of multi-view images.

In this paper, we propose that 3D Gaussians are a useful learned mid-level image representation due to their non-uniformity properties. We take advantage of the splatting operation from (Kerbl et al., 2023) that enables end-to-end training of mid-level representations with image-based losses. We then use Gaussian primitives in a representation learning framework rather than a single-scene optimization-based 3D reconstruction as in (Kerbl et al., 2023), thus opening the door for using learned 3D Gaussian representations in computer vision applications.

The key advantages of Gaussian splatting include its ability to adapt to scene complexity, efficient rendering, and high-quality reconstructions. Unlike uniform voxel grids, Gaussian primitives can vary in size and density, allowing for more compact and expressive representations of 3D scenes. This adaptability makes them particularly suitable for a wide range of computer vision tasks, from 3D reconstruction to novel view synthesis and beyond.

## 3 METHOD

We propose a method that reconciles pixel-based learning, the mainstream in self-supervised learning, and latent-based learning, which can impose extra properties on representations. Our key insight is that end-to-end learnable 3D Gaussians are good candidates for mid-level image representations due to their non-uniform properties. Given a large collection of images, we train a Masked Autoencoder (MAE) (He et al., 2022) to reconstruct full images from their masked inputs. The MAE encoder is a ViT (Dosovitskiy et al., 2020) that learns to encode the visible square patches of the masked images into learned embeddings. However, rather than predicting patches of pixels directly as in MAE (He et al., 2022), our ViT-based decoder predicts explicit 3D Gaussians (Kerbl et al., 2023) — their color, 3D center location, scale, and orientation. We then render these Gaussians as images with a splatting differentiable renderer and train the entire model using an MSE loss in pixel space.

### 3.1 PRELIMINARIES

**Self-supervised masking autoencoders.** Masking autoencoders model the data distribution by randomly masking parts of the data and predicting the masked parts. In language, BERT (Devlin et al., 2018) is trained by masking part of the text tokens and predicting them using a transformer model (Vaswani et al., 2017). In vision, MAE (He et al., 2022) and BEiT (Bao et al., 2021) mask image patches at the input and predict the masked regions. In MAE (He et al., 2022), a ViT (Dosovitskiy et al., 2020) encoder encodes the visible patches, and a smaller decoder ViT model is used with masked tokens to reconstruct the masked patches.

**3D Gaussian primitives and splatting.** Our model learns a mid-level image representation using the 3D Gaussian primitives originally proposed for optimization-based single-scene 3D reconstruction by (Kerbl et al., 2023). Each Gaussian is characterized by a 3D covariance matrix $\Sigma \in \mathcal{R}^{3 \times 3}$ and a center location $p \in \mathcal{R}^3$. Additionally, each Gaussian is assigned a color $r \in \mathcal{R}^3$ and an opacity $o \in \mathcal{R}$ to encode the scene content. For image rendering, these Gaussian primitives are transformed into camera coordinates and projected onto the image plane using volume splatting. Due to the differentiable nature of this process, the attributes of the Gaussian primitives can receive gradients from the rendered image. In our work, we adopt the standard approach of factorizing the covariance matrix $\Sigma = RSS^T R^T$ into a scaling matrix $S = \text{diag}(s) \in \mathcal{R}^{3 \times 3}$ represented by a scale vector $s \in \mathcal{R}^3$, and a rotation matrix $R \in \mathcal{R}^{3 \times 3}$ represented by a rotation quaternion $\phi \in \mathcal{R}^4$. Consequently, each Gaussian is parameterized by a 14-dimensional vector $g = \{p, s, \phi, r, o\} \in \mathcal{R}^{14}$.

### 3.2 OUR APPROACH

Our model has a ViT-based encoder model, a lightweight decoder model, and a differentiable renderer. Fig. 2 shows a high-level overview of our method. For a given image, we first patchify it into $N$ patches and randomly mask them with a masking ratio $r$, resulting in $n$ visible patches. The ViT encoder model only sees the visible patches and encodes them from patches to latent embeddings, $x_i \in \mathcal{R}^{d_{enc}}$, $i \in \{1, 2, 3, ...n\}$.

Assume the decoder has $k$ learnable query tokens $q_j \in \mathcal{R}^{d_{dec}}, j \in \{0, 1, 2, ...k\}$. Note that $k$ can be any value irrespective of the number of masked tokens. We project the encoder latent to $\hat{x}_i \in \mathcal{R}^{d_{dec}}$ and concatenate it with the query tokens.

$$X_{dec} = \{\hat{x}_i\}_{i=1}^n \cup \{q_j\}_{j=1}^k \tag{1}$$

The decoder sees the $X_{dec}$ tokens and predicts $k$ Gaussians, one for each query token (we discard the predictions for the latent tokens). Each Gaussian is parameterized by a 14-dimensional vector $g_j = \{p, s, \phi, r, o\} \in \mathcal{R}^{14}$.

Once we have $k$ predicted Gaussians, we splat them on a plane with a fixed camera projection and render the splatted Gaussians to generate an image. We limit the size of the Gaussians by using an effective scale $c \cdot \texttt{sigmoid}(s)$. Here, $c$ controls a Gaussian's maximum size. After rendering, we use a mean squared error loss to compare the reconstructed image with the input image on the originally masked pixels.

Note that since the Gaussians are the output of the decoder, they are effectively randomly initialized. This is in contrast to the typical usages of Gaussian splatting for 3D reconstruction that rely on point-cloud initialization. In this work, we do not use any prior knowledge. We directly learn all the Gaussian properties from reconstructing the image.

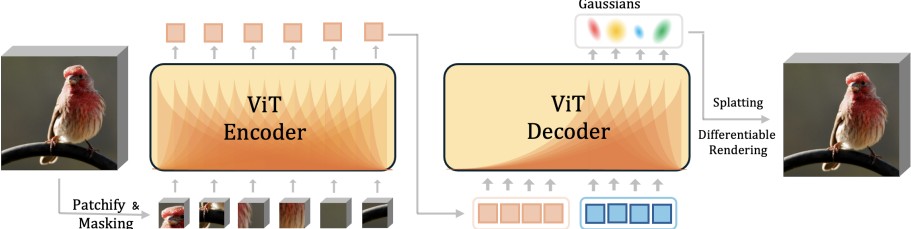

Figure 2: **Masked Autoencoding via Gaussian Splatting:** The ViT Encoder processes masked input image patches to produce latent embeddings. The ViT Decoder then predicts explicit Gaussian parameters based on query tokens, including color, opacity, center, scale, and orientation. These Gaussians are then rendered via differentiable volume splatting (Kerbl et al., 2023) to reconstruct the original image. We pre-train our models fully end-to-end with self-supervision.

## 4 EXPERIMENTS

### 4.1 DESIGN CHOICES

First, we will explore various design spaces for pre-training our models. All our experiments in this section are based on a ViT-base encoder and a lightweight decoder and measured by ImageNet (Deng et al., 2009) classification performance. All the models are trained for 400 epochs. We use a base learning rate of $1e-4$ with cosine decay with AdamW (Loshchilov & Hutter, 2017) optimizer. We evaluate our pre-trained models using linear probing and full finetuning.

**Number of Gaussians:** Unlike MAE, our decoder model is fully decoupled from encoder tokens. Therefore, we can use any number of Gaussians for decoding. We train 4 models that learn to decode 64, 128, 256, and 512 Gaussians, respectively. Fig. 3 shows ImageNet classification performance under linear probing and full fine-tuning as a function of the number of Gaussians. With linear probing, performance monotonically increases as we increase the number of Gaussians. With full fine-tuning, we see a similar behavior at first, but it saturates after 256 Gaussians.

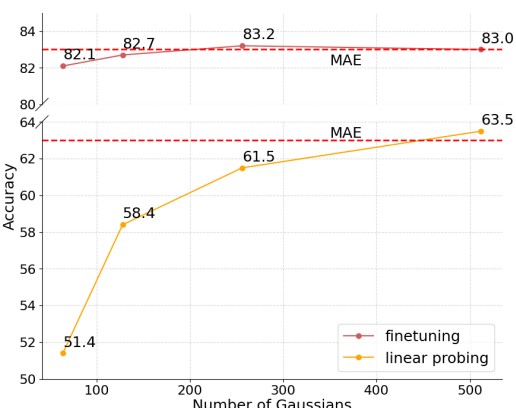

Figure 3: **Number of Gaussians:** ImageNet classification performance with 64, 128, 256, and 512 Gaussians at the decoder during pre-training. We evaluate these models on linear probing and full finetuning. As we increase the number of Gaussians, the performance with linear probing increases monotonically. For full fine-tuning, we see similar behavior at first that saturates after 256 Gaussians.

**Gaussian Scale:** Since the Gaussians are predictions from the decoder, we can not control their initialization explicitly. We only have an activation function after decoder predictions: for x,y,z we use tanh, and for scale and quaternions we use sigmoid. Since we are learning the scales from randomly initialized Gaussians, we limit the Gaussians from being too big by passing them through the scaled sigmoid function before rendering ($c \times \texttt{sigmoid}(scale)$). Here, we study the effect of $c$ on the representation quality. Table 1a shows how classification performance on ImageNet changes by varying the maximum allowed scale values. This essentially controls how big each Gaussian can be and how many pixels they can influence. This variable has only a small effect on the classification performance. However, small scale values greatly hinder the reconstruction quality; see Fig. 4 for qualitative differences.

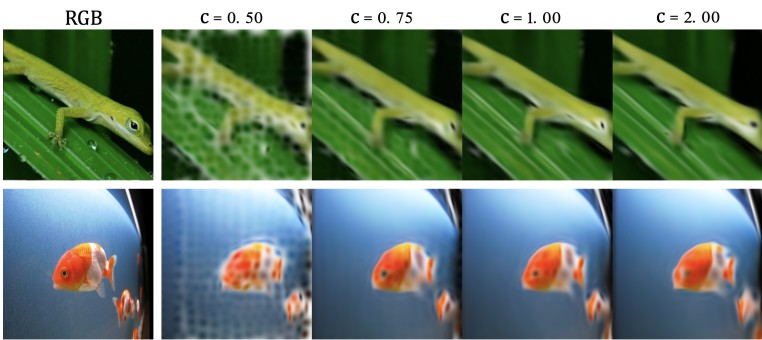

RGB   c = 0.50   c = 0.75   c = 1.00   c = 2.00

Figure 4: **Effect of scale on reconstruction:** Here we have visualizations to show that With small-scale Gaussians the model can not complete the whole image with a fixed number of Gaussians.

| Scale Factor | ft | lin | | Masking | ft | lin | | Loss | ft | lin |
|---|---|---|---|---|---|---|---|---|---|---|
| 0.50 | 82.7 | 61.1 | | 0.70 | 82.6 | 60.2 | | All | 81.8 | 57.4 |
| 1.00 | **83.2** | **61.9** | | 0.75 | **83.2** | **61.9** | | Masked | **83.2** | **61.9** |
| 2.00 | 83.1 | 60.9 | | 0.80 | 82.4 | 61.9 | | Masked (norm) | 76.9 | 31.7 |

| (a) Max scale of Gaussians ($c$) | (b) Masking Ratio | (c) Loss patches |
|---|---|---|

Table 1: **Ablation studies on different model space:** We ablate various configurations for pre-training, including **a)** the maximum allowed scale for a Gaussian, **b)** masking ratio at the encoder, and **c)** which loss is used for pre-training, from applying loss on all patches, only on masked patches to patch-normalized loss. We report classification performance on ImageNet, where higher is better.

**Masking ratio:** We study the dependence of our pre-trained models on the masking ratio in Table 1b. MAE (He et al., 2022) showed that higher masking rates allow the model to learn better representations and make the training much more efficient. Using differentiable rendering instead of transformer-based rendering did not change this behavior.

**Loss:** We also study how to apply the loss on patches. Should we apply on masked patches or all the patches? On pixel prediction or normalized pixel prediction? MAE (He et al., 2022) benefited from predicting normalized patches when applying the loss on masked patches. In our case, a normalized patch loss significantly hurts model performance, as shown in Table 1c. This is due to the fact that all the Gaussians can influence a specific pixel value, a constraint that makes it harder to reconstruct locally normalized patches. Finally, we have a similar observation as in MAE (He et al., 2022); when the loss is applied only to the masked tokens, the model performs slightly better than the model trained with loss applied to all the patches.

**Final model:** Based on our findings from the above experiments, our final model is a ViT-base model that is pre-trained with a 0.75 masking ratio for 400 epochs, with patch loss applied to the masked tokens and has 512 Gaussians at the decoder during per-training. We will use this model for the rest of our experiments. All our models are pre-trained on the ImageNet (Deng et al., 2009) 1k dataset.

## 4.2 SUPERVISED TASKS

To measure the representation quality of our pre-trained models, we evaluate our models on ImageNet-1k (Deng et al., 2009) classification and COCO (Lin et al., 2014) object detection. We use the ViT-base model, trained for 400 epochs with a masking ratio of 0.75, for fine-tuning on ImageNet and COCO.

**Image Recognition:** Table 2a shows the performance of various pre-trained models on ImageNet classification top-1 accuracy. We fine-tuned our model for 100 epochs, following the protocols of MAE (He et al., 2022). While achieving comparable performance to MAE, our models show an interesting trend when increasing the number of Gaussians, as in Fig. 3. Scaling this further without increasing compute requirements would necessitate further modifications, which we leave for future directions.

**Object Detection and Segmentation:** As another important way to evaluate our encoder representations, we transfer the representations learned with our pipeline via fine-tuning, to object detection and segmentation. We follow the protocol of ViTDet (Li et al., 2022) and evaluate on the COCO

| Method | Top-1 |
|---|---|
| *Discriminative Approaches* | |
| SimCLR | 76.5 |
| BYOL | 79.6 |
| DINO | 82.8 |
| DINOv2 | 88.3 |
| *Generative Approaches* | |
| BEiT | 83.2 |
| MAE (1600-ep) | 83.6 |
| MAE (400-ep) | 83.0 |
| GMAE (400-ep) | 83.2 |

(a) ImageNet classification.

| Method | $AP^{box}$ | $AP^{mask}$ |
|---|---|---|
| Supervised | 47.6 | 42.4 |
| MAE (1600-ep) | 51.8 | 44.9 |
| MAE (400-ep) | 50.6 | 45.0 |
| GMAE (400-ep) | 50.2 | 44.5 |

(b) COCO object detection and segmentation.

Table 2: **GMAE performs comparably to MAE (He et al., 2022)** on supervised tasks.

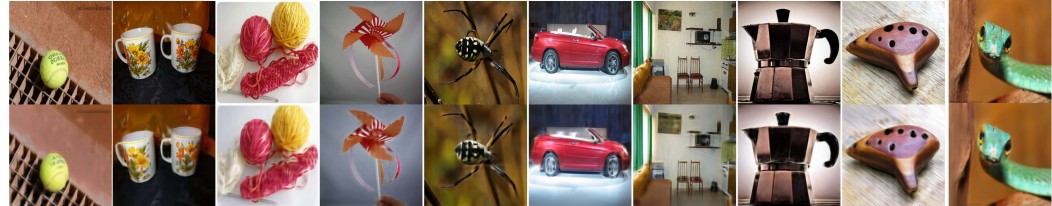

Figure 5: **Reconstruction Quality:** Examples of test-time reconstructions when the input is fully visible (mask ratio=0). The first row is the RGB images and the second row is our reconstructions. Having a decoupled decoder allows us to perform inference with any masking ratio, even though this model is trained with a 0.75 ratio. The dynamically learned non-uniform spatial and scale distribution of Gaussians enables GMAE to reconstruct high-frequency regions like lines and edges. Our rFID score is 89.45, while MAE rFID is 98.12 (smaller is better), and PSNR of GMAE is 18.74 and of MAE is 18.63.

benchmark (Lin et al., 2014). Again, we find GMAE performs similarly to MAE in $AP^{box}$ and $AP^{mask}$, and significantly outperforms supervised pre-training. See Table 2b.

## 4.3 UNSUPERVISED TASKS

In this section, we study the properties of the decoder of our pre-trained models. Unlike MAE, we now have access to an intermediate representation that we can edit and modify *without* re-training. Given an input image, we fully encode the image and generate Gaussians that we splat to reconstruct the image. First, we evaluate the image reconstruction quality of our model. On the ImageNet validation set, our ViT-base model achieved $89.45$ on reconstruction FID, while the MAE ViT-base model got $98.12$ (lower is better). This improvement in reconstruction quality is due to the non-uniformity of the learned distribution of Gaussians, which allows GMAE to model high-frequency information, as shown in Fig. 5. As a result, our reconstructions can be used directly for other tasks without needing to add a GAN loss or an upsampling layer on top.

Another advantage of using a 3D Gaussian representation to represent 2D images is that it learns to separate objects in the $z$ direction. This may be due to the fact that with random initialization, the points closer to the camera represent low-frequency information, while the points far from the camera model the high-frequency information (see Section 4.4). To segment an image along the $z$ axis, we simply sort the predicted Gaussians based on their depth value and group them into $L = l_0, l_1, ... l_d$ groups. To split an image into two layers, we try to render it as two images using $l_0, ..., l_n$ and $l_0, ..., l_n, l_{n+1}$. If the difference at a pixel is larger than a specific threshold $th$, we assign that pixel to the layer $n + 1$. Fig. 7 shows the layering effect of a Gaussian representation with 64 layers.

**Zero-shot Figure-Ground Separation:** The layer-wise rendering allows us to perform figure-ground segmentation for free. We simply get the layer-wise rendered image and apply a threshold to obtain foreground-background segmentation. We evaluate our approach on the PASCAL dataset (Everingham

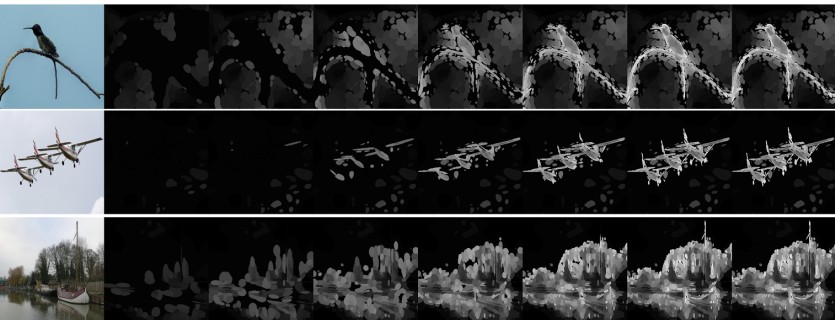

Figure 6: **Visualization of Gaussian layers:** The first column shows the RGB input image, and the subsequent columns show the Gaussian layers in inverse-depth ordering. The layer-wise rendering highlights the model's ability to separate objects and represent them in distinct frequency layers, enabling zero-shot foreground-background segmentation and edge detection.

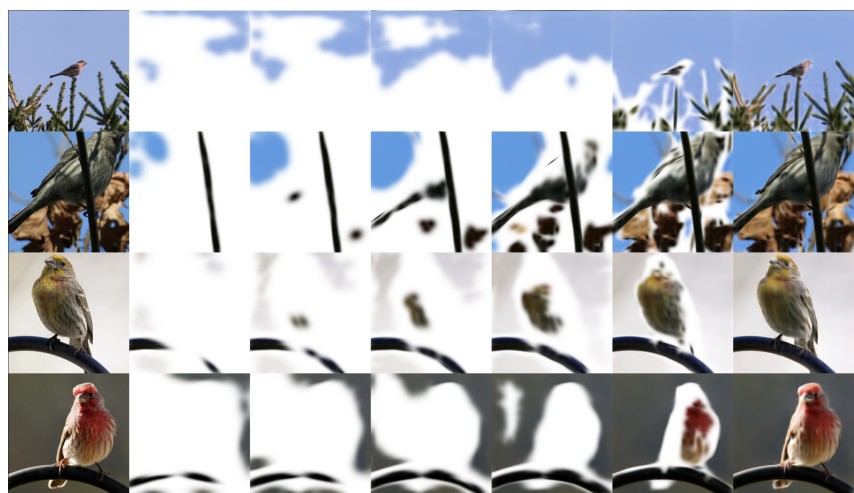

Figure 7: **Visualization of Gaussian layers on RGB:** The first column shows the RGB input image, and the rest of the columns show rendering of partial depth Gaussians. We select first $K$ Gaussians from all the Gaussians after sorting based on depth. Then we only render these $K$ Gaussians. The Figure shows $K = 32, 64, 128, 256, 512, 1024$, and the last being fully rendered image using all Gaussians.

et al., 2015) on foreground segmentation and single object detection tasks as in (Bar et al., 2022) (see Table 3). The "copy" baseline (Bar et al., 2022) predicts a random ground truth mask from the training set. We note that GMAE performs better than other few-shot baselines despite being a zero-shot approach.

**Zero-shot Edge detection:** In a similar fashion, as in zero-shot figure-ground segmentation, we can simply take the layer-wise rendered image and find edges with a discontinuity in the $z$ direction. Additionally, the number of layers determines the granularity in the edge detection; for example, a large number of layers means we will detect more fine-grained edges. Fig. 9 shows the detected edges of our zero-shot method with varying number of layers. We also quantitatively evaluate our method on BSDS500 (Arbelaez et al., 2010). Table 4 shows that our method archives reasonable performance for zero-shot detection without ever being trained for segmentation or detection. Fig. 8 shows that as we decrease the number of layers (*e.g.*, increasing the width of each layer), the quality of edge detection gets better, allowing us to have a hierarchy of edges.

| Model | Foreground Segmentation ↑ | | | | Single Object Detection ↑ |
|---|---|---|---|---|---|
| | Split 0 | Split 1 | Split 2 | Split 3 | |
| Copy (Bar et al., 2022) | 12.92 | 17.90 | 13.52 | 15.29 | 13.50 |
| BEiT (Bao et al., 2021) | 0.38 | 0.93 | 0.90 | 0.95 | 0.32 |
| VQGAN (Esser et al., 2020) | 6.96 | 10.55 | 9.59 | 9.43 | 4.99 |
| MAE (He et al., 2022) | 1.92 | 6.76 | 3.85 | 4.57 | 1.98 |
| MAE-VQGAN (Bar et al., 2022) | 2.22 | 7.07 | 5.48 | 6.28 | 3.21 |
| GMAE | 17.85 | 19.09 | 19.16 | 16.55 | 20.26 |

Table 3: **Zero shot segmentation and detection:** We evaluate on a random subset of the PASCAL (Everingham et al., 2015) dataset for figure-ground segmentation and single object detection.

| Method | ODS | OIS | AP |
|---|---|---|---|
| HED | 0.788 | 0.808 | 0.840 |
| EDETR | 0.840 | 0.858 | 0.896 |
| *Zero-shot:* | | | |
| Sobel filter | 0.539 | - | - |
| Canny | 0.600 | 0.640 | 0.580 |
| Felz-Hutt | 0.610 | 0.640 | 0.560 |
| SAM | 0.768 | 0.786 | 0.794 |
| GMAE | 0.515 | 0.524 | 0.248 |

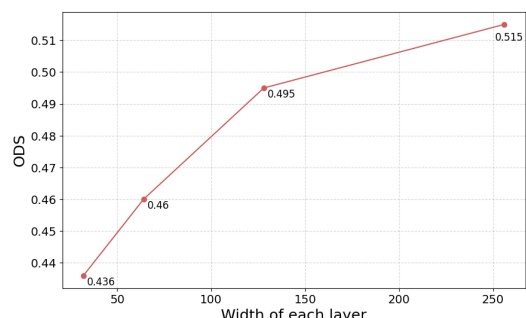

Table 4: **Zero-shot transfer to edge detection:** on BSDS500 (Arbelaez et al., 2010).

Figure 8: **Edge Detection:** quality improves as we increase the width of each layer.

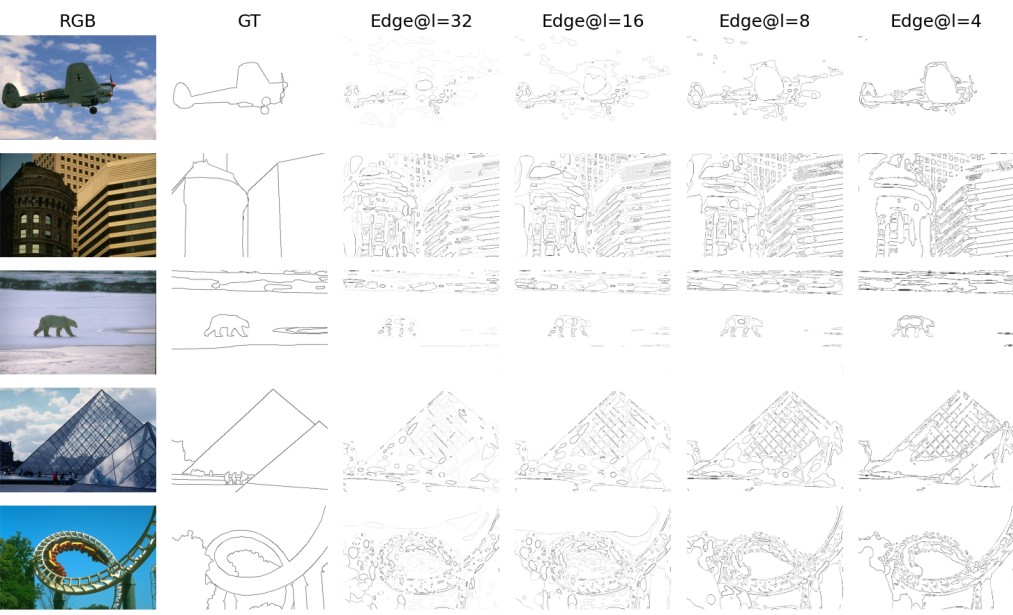

Figure 9: **Zero-shot edge detection:** Gaussians from our decoder are grouped into equal sets of layers, based on their depth ordering. Then we render each layer one by one and update the pixel if adding that layer makes significant change in that pixel. From this layered image, we simply find edges if there is a discontinuity from one layer to another. We can get a hierarchy of edges by decreasing the number of layers (or increasing the width of a layer). These results are on BSDS500 (Arbelaez et al., 2010). $l$@16 means we have a total of 16 uniformly grouped layers.

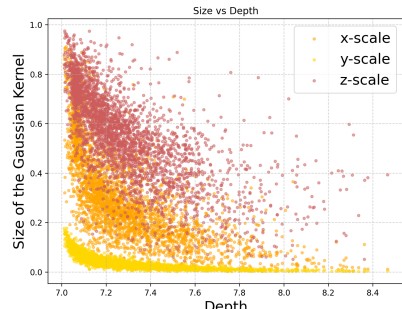

Figure 11: **Distribution of Gaussians on xy-plane:** We plot the xy position of Gaussian centers along with their opacity values. Unlike patches, Gaussians are positioned dynamically based on the image. For example, in the 4th image, Gaussians are arranged in a circular pattern, while in the 2nd image there arranged as a rectangle. This degree of freedom allows them to add high-frequency signals to the image, by concentrating more Gaussians to those regions.

## 4.4 QUALITATIVE RESULTS

**Distribution of Gaussians in xy:** Fig. 11 shows how the Gaussians are arranged differently in space for different images. In patch-based methods such as MAE, patches are arranged in a uniform tile. Even if we decrease the patch size to increase the number of patches, they will still be uniformly allocated across all regions. However, in our work, a Gaussian can go to any part of the image, which allows them to dynamically position themselves based on the input image. This property allows modeling high-frequency regions with high fidelity. As shown in Fig. 5, our reconstructions can capture high-frequency regions such as faces and intricate patterns.

**Size vs Depth:** Fig. 10 shows a clear trend: the Gaussians with larger scale values lie closer to the camera, while the ones with smaller scale values, lie away from the camera, on average. This distribution validates our previous hypothesis in Sec 4.3, that low-frequency blobs lie closer and cover larger regions, while high-frequency blobs lie away from the camera, on average. At the start, some Gaussians will be closer to the camera, influencing more pixels, therefore representing low-frequency regions. Our layering results are the outcome of this property. In the real world, backgrounds tend to have low-frequency regions while objects usually have high-frequency details. This correlation leads to our zero-shot results.

Figure 10: **Size vs. Depth:** Distribution of factorized scale values ($s$) over predicted depth.

## 5 DISCUSSION

This paper presents GMAE, a self-supervised image representation learning approach that extends MAE to include a learned intermediate Gaussian representation. We show that learning to represent images with 3D Gaussians has several built-in advantages that stem from their non-uniform dynamical allocation of scale, location, and distribution. Our method, therefore, lends itself to zero-shot capabilities such as foreground-background segmentation, image layering, and edge detection. Along with these advantages, we demonstrate that the representation learned by our method is on par with MAE on standard supervised image recognition tasks and that it transfers to downstream tasks such as detection and segmentation via fine-tuning.

GMAE still exhibits several empirical limitations. For example, setting larger scale values at the start of training results in a more challenging optimization. Compared to the number of Gaussians typically used for 3D reconstructions (up to millions), the number of Gaussians we have used in GMAE is bottlenecked by compute, and increasing it to more than a thousand can cause major slow-downs for pre-training. An interesting future direction is to further accelerate our pipeline.

We hope our exploration can inspire more work in this direction and unlock the next generation of techniques that effectively model visual data.

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
