## A  TRAINING DETAILS

Here we provide configurations used for training 5 and finetuning 6 our models. For pre-training, we follow a similar recipe as in MAE (He et al., 2022). For linear probing we use AdamW instead of LARS (You et al., 2017) as in MAE (He et al., 2022).

| Config | Value |
| --- | --- |
| optimizer | AdamW (Loshchilov & Hutter, 2017) |
| base learning rate | 1e-4 |
| minimum absolute lr | 0.0 |
| weight decay | 0.05 |
| optimizer momentum | $\beta_1 = 0.9, \beta_2 = 0.95$ |
| batch size | 4096 |
| learning rate schedule | cosine decay (Loshchilov & Hutter, 2017) |
| warmup epochs | 20 |
| training epochs | 400 |
| augmentation | RandAug (9, 0.5) |
| label smoothing | 0.1 |
| mixup | 0.0 |
| cutmix | 0.0 |
| drop path | 0 |
| layer decay | 1.0 |

Table 5: **Pre-training recipe for GMAE**.

| Config | Value |
| --- | --- |
| optimizer | AdamW (Loshchilov & Hutter, 2017) |
| base learning rate | 1e-4 |
| minimum absolute lr | 0.0 |
| weight decay | 0.01 |
| optimizer momentum | $\beta_1 = 0.9, \beta_2 = 0.999$ |
| batch size | 4096 |
| learning rate schedule | linear decay |
| warmup epochs | 10 |
| training epochs | 90 |
| augmentation | BasicAug |
| label smoothing | 0.1 |
| mixup | 0.8 |
| cutmix | 0.5 |
| drop path | 0.2 |
| layer decay | 0.8 |

Table 6: **Fine tuning recipe GMAE**.

## A.1 MORE SAMPLES

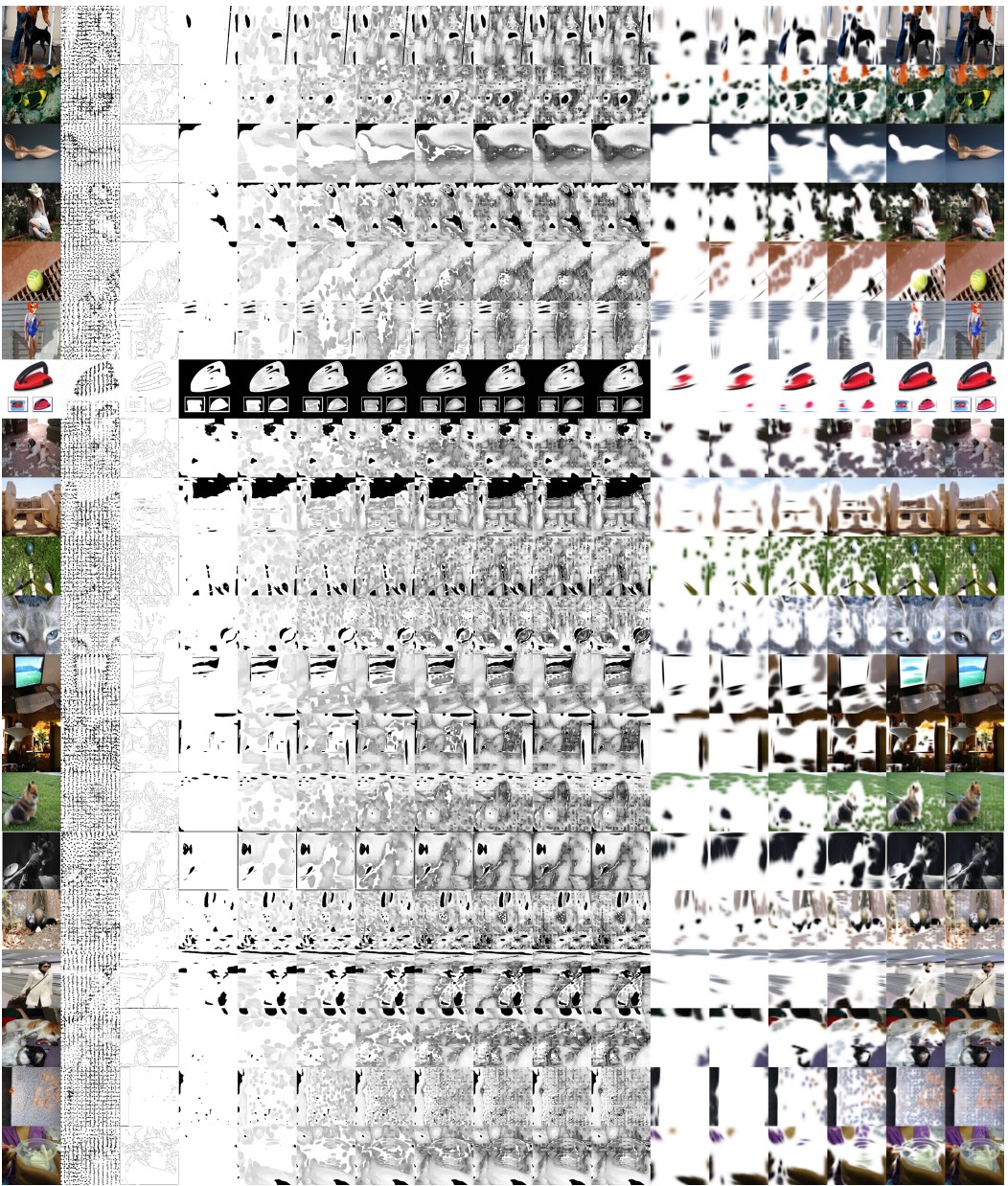

Figure 12: **Unfiltered Samples:** We show more samples from ImageNet-1K, which RGB, XY positions of Gaussians, layering of Gaussians, and RGB layering.

## A.2  RESIDUAL GAUSSIANS

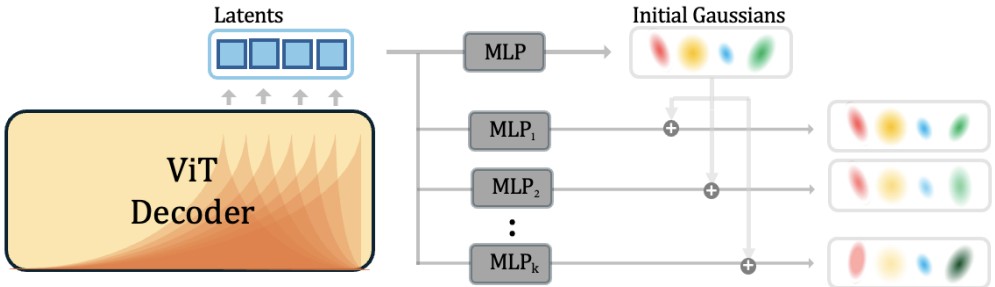

Figure 13: **Learning Residual Gaussians:** After we pretrained the model on $N$ number of Gaussians, we increase the total number of Gaussians by learning $k$ different small changes to the initial Gaussians. We learn $k$ mlp layers, which learns a small changes to the initial Gaussians. We initialize $k$ mlp-layers with zero weights. Finally all Gaussians from each mlp head is is combined to get a total of $k \times N$ Gaussians.