# OpenReview forum: "Gaussian Masked Autoencoders"
_ICLR.cc/2025/Conference — Submitted to ICLR 2025_

### Official Review · Reviewer_PMrf · 2024-10-27

**Soundness:** 2
**Presentation:** 2
**Contribution:** 2
**Rating:** 5
**Confidence:** 4

**Summary:**

The paper proposes a new self-supervised learning method called Gaussian Masked Autoencoder (GMAE) which combines the advantages of pixel-level and intermediate representation learning. It uses 3D Gaussian distributions as intermediate representations to capture richer image information and improve image generation and processing abilities. GMAE achieves comparable performance to supervised learning while significantly improving zero-shot learning tasks such as segmentation, layering, and edge detection. The paper also demonstrates the potential of GMAE in downstream tasks like visual recognition and object detection.

**Strengths:**

1. The paper presents a novel approach to self-supervised learning by introducing Gaussian Masked Autoencoder (GMAE) that utilizes 3D Gaussian distributions as intermediate representations. This idea is different from traditional methods that use pixel-level reconstruction, and the authors demonstrate the effectiveness of this approach through experiments.

2. The paper is well-written and easy to understand. The authors provide details about the implementation and evaluation metrics, making it possible to replicate the results. The empirical results presented in the paper are convincing and support the claims made by the authors.

**Weaknesses:**

1. Lack of comparison to state-of-the-art methods: The paper does not compare the proposed method to other existing methods for self-supervised learning, such as contrastive learning or clustering-based methods. This makes it difficult to assess the relative performance of GMAE compared to other approaches.

2. I didn't understand the immediate motivation for choosing Gaussian splatting to enhance MAE in this paper. What qualities does Gaussian splatting have to help MAE? Can Gaussian splatting be replaced with NeRF or other 3D representations?

**Questions:**

See Weaknesses. I will change my rating based on the responses.

---

> ### Author Response · Authors · 2024-11-22
>
> We thank the reviewer for comments on our paper where the “idea is novel”, presented in a “well written paper and wide range of experiments which are reproducible”. Here we answer the questions the reviewer is asking, and we are happy to answer more questions or run more experiments for the rebuttal. We will also open source the code and the models.
>
> **Comparison with contrastive learning methods.**
>
> We thank the reviewer for the suggestion. We will add more baseline contrastive approaches in Table 2a. We will also explore more tasks. For example, we have already added a depth estimation task, and we show GMAE performs better than DINOv2 in  depth estimation. We want to highlight that, even after zero-shot capabilities were introduced, the model does not lose its encoder representation power.
>
>
> **“What qualities does Gaussian splatting have to help MAE? Can Gaussian splatting be replaced with NeRF or other 3D representations?”**
>
> We were interested in learning from large scale images, and MAE is proven to be the best approach to learn good **semantic** representations. On the other hand, Gaussians are good way to learn **structure/geometry**. A combination of these two approaches allowed us to learn good semantic representations (as shown in imagenet, coco experiments), as well as some geometry based on 2.1d representations (layering and edge detections).
>
> Regarding NERF vs Gaussians, the more important property we see from Gaussian Splatting is that it's an unstructured sparse representation that automatically allocates more resources on finer details. Neither of the NeRF variants come with this property. Though some other 3D representations such as point cloud or meshes also allow spatially variant sparsity, they are much harder to optimize than 3D Gaussians. So we see Gaussians are a great way to represent not only 3D content, but also (layered) 2D content in the machine learning context.

---

### Official Review · Reviewer_Zk2V · 2024-10-30

**Soundness:** 3
**Presentation:** 3
**Contribution:** 3
**Rating:** 6
**Confidence:** 3

**Summary:**

In this paper, the authors present a self-supervised image representation learning method to extend Masked Autoencoders with 3D Gaussian Splatting framework. The general framework is a ViT based auto-encoder which takes masked patches from given images as input. The key idea is that instead of predicting image patches, the ViT based decoder regresses the parameters of 3D Gaussians for further rendering. To validate the importance of this technical upgrade, the authors perform comparisons on both supervised tasks and unsupervised tasks as well as ablation studies on different training mechanisms.

**Strengths:**

+ Generally, the paper is well written. I can understand it easily.
+ As for me, the idea of applying learned 3D Gaussians as primitives for downstream tasks in unsupervised learning is novel and critical to the computer vision community.
+ The authors perform evaluation for both supervised and unsupervised tasks to show the empirical significance of their 3DGS based network upgrades.

**Weaknesses:**

+ The evaluation datasets and used baselines seem to be a bit outdated. The latest baselines (MAE and MAE-VQGAN) were published in 2022 while the latest testset (PASCAL) was published in 2015. Could the authors evaluate their method on some datasets listed in Figure 8 of SAM [1] with modern large-scale unsupervised learning methods? For example, datasets like COCO-Stuff or ADE20K? And baselines like SAM or DINO v2? Or other related datasets and baselines?
+ There are no failure case examples to justify the possible future work of GMAE method. Ideally, the failure cases might reveal limitations in the Gaussian representation and highlight scenarios where the method struggles compared to pixel-based approaches.
+ Some typos which include:
1. L313, "the ViT base model" --> "the ViT based model";
2. L533, the "For example," are repeated twice.

[1] Segment Anything, ICCV 2023

**Questions:**

+ As mentioned in L200 and L227, how to get the query tokens? What these tokens could be?

---

> ### Author Response · Authors · 2024-11-22
>
> Official Response by Author for the Reviewer Zk2V
>
> We thank the reviewer for comments that mention that our “idea is novel and critical to the computer vision community”, and that acknowledge our well written paper and wide range of experiments. Here we answer the questions the reviewer is asking, and we are happy to answer more questions or run more experiments for the rebuttal and we fix the typos in the paper.
>
>
> **“The evaluation datasets and used baselines seem to be a bit outdated”**
>
> For edge detection, even SAM is evaluated on the BSD500 dataset, and for the segmentation tasks, we are following the protocol and the code in MAE-VQGAN. We have added new results on NYU-scenes on depth estimation. We appreciate the suggestion to compare to DINOv2 and added it as an additional baseline.  We will add more tasks which are relevant for this during this discussion period.
>
> | Model   | RMSE   |
> |---------|--------|
> | DINOv2  | 0.4761 |
> | MAE       | 0.4378  |
> | GMAE    | 0.4336 |
>
>
> **“There are no failure case examples to justify the possible future work of GMAE method.”**
>
> Thanks for bringing up this point. We did share unfiltered, randomly chosen samples from our models, in appendix figure 12. A few failure modes we would like to list are a) in the cases of layering, sometimes the layers are still not fully disentangled from colors. b) another limitation of this work is the use of L2 loss, which tends to give results which are blurry, compared to, for example, diffusion style loss. c) our model suffers from a feedforward vs optimization tradeoff, optimization approach like gsplat training would give better results while being slow, while gmae on other hard, being a feedforward fast approach, but giving not the best reconstruction results. d) the number of gaussians was another main limitation of our work, however, based on the new results, we have fixed this by learning additional residual gaussians.  Now we can learn up to 4096 gaussians.
>
> “How to get the query tokens? What could these tokens be?”
> Query tokens are learned tokens, which were trained end-to-end during the training of the encoder and the decoder, and we initialize them randomly with zero mean and 0.02 variance. At test time, The query tokens and the latent tokens from the encoder are concatenated and passed through the decoder, and the query tokens are projected into gaussians for rendering. We will make this point more clear in the updated manuscript.

---

> > ### Comment · Reviewer_Zk2V · 2024-11-24
> > **Thank you for your response**
> >
> > I thank the authors for their response. Now my concerns about failure cases and expositions are addressed. But my concerns about the evaluation protocols still remains. I think results on more complicated datasets like COCO-Stuff or ADE20K and with more complex tasks like segmentation could better reveal the effectiveness of GMAE. But as an initial attempt, I also do not want to be too harsh. Given that I am not an expert in this area, I would stay slightly positive towards this submission but will not strongly champion it.

---

> > > ### Author Response · Authors · 2024-11-24
> > >
> > > We thank the reviewers for their feedback and thanks for their suggestions. We are still working on getting results  on COCO-stuff and ADE20K. We will try our best to get these results before the end of the discussion period.

---

### Official Review · Reviewer_PWCZ · 2024-11-03

**Soundness:** 3
**Presentation:** 3
**Contribution:** 3
**Rating:** 6
**Confidence:** 4

**Summary:**

This paper introduces GMAE, a method that integrates MAE with Gaussian Splatting. It claims that GMAE is a better way to represent mid-level features. Instead of reconstructing masked pixels, GMAE predicts a set of Gaussians, each parameterized by a 14-dimensional vector. The authors have meticulously designed the model, and some observations align closely with those of MAE. Empirical results demonstrate that GMAE achieves performance comparable to MAE on supervised tasks and exhibits satisfactory zero-shot capabilities in unsupervised tasks.

**Strengths:**

- The idea looks intriguing to me. The non-uniformity inherent in Gaussian representation distinguishes it from traditional patch-based methods.
- Experimental results are comprehensive and convincing.
- The numerous visualizations offer an intuitive grasp of the methodology and outcomes.
- The paper is well-written and easy to follow.

**Weaknesses:**

- In supervised tasks, the model primarily utilizes the ViT encoder, without incorporating Gaussian representations. The effectiveness of Gaussian representations is demonstrated in unsupervised tasks. Demonstrating a positive impact on image generation would significantly enhance the paper’s contributions.
- The limited number of Gaussians employed constrains the model’s reconstruction capabilities for image generation. If increasing the Gaussian count presents a bottleneck, this limitation could hinder its application in image generation tasks.

Overall, my main concern lies in the scalability of the method. But as an initial attempt, I think the paper is above the acceptance threshold .

**Questions:**

- In line 089, the statement “the addition of splatting increases compute time by 1.5%” would be more informative if the authors provided the absolute compute times for both MAE and GMAE, facilitating a clearer comparison.
- In Figure 7, arranging the visualization of Gaussian layers from shallow to deep depths could be more intuitive, as objects closer to the camera often hold greater significance.

---

> ### Author Response · Authors · 2024-11-22
>
> We thank the reviewer for comments on our intriguing idea, meticulously designed model, and satisfactory zero-shot capabilities. Here we answer the questions the reviewer is asking, and we are happy to answer more questions or run more experiments for the rebuttal.
>
> **“Demonstrating a positive impact on image generation would significantly enhance the paper’s contributions.”**
>
> We agree! However, our current goal is to show the representation capabilities of the self-supervised models rather than their generation capabilities, which would require re-thinking the model, for example, using DiT models. While this is not the scope of the paper, to archive this number of gaussians was one of the main bottlenecks, which the reviewer also mentioned. Based on the new results we have fixed this problem and were able to scale up to 4096 gaussians. This could be a potential tokenizer to train generative models now, however since that is beyond the scope of this work, we will leave it for future work.
>
> **“If increasing the Gaussian count presents a bottleneck, this limitation could hinder its application in image generation tasks.”**
>
>
> We agree with the reviewer, that the limit on  the number of gaussians is a limitation in the current model, and the current design we are limited by memory. We tried to solve this problem by learning more gaussians step by step, as residual gaussians. First, we pretrained the model with 256 Gaussians, and after that, we initialize an k MLP layers, to project changes in 14 features to k*256 Gaussians. These mlp layers essentially learn the small changes from the main 256 gaussians. At the start these mlp layers are initialized with zero weights, hence they don't affect the reconstruction for the original 256 gaussians.
>
> With this setting, we were able to train up to 4096 gaussians, but not limited (we will update here with more gaussians), since it is not limited by memory anymore. Below we show the reconstruction FID as we increase the number of gaussians step by step.  Finally, we also fine tune the model (4096*) without masking and on full reconstruction, and it achieves 18 rfid, without perceptual loss or vae.
>
> | Number of Gaussians   | rFID   |
> |---------|--------|
> | 256         |  89.45  |
> | 1024       |  80.32  |
> | 4096       |  63.87  |
> | 4096*      |  18.71  |
>
>
> **“Would be more informative if the authors provided the absolute compute times for both MAE and GMAE”**
>
> For MAE training on V100 gpus, takes on average time: 0.6471 seconds and Standard deviation: 0.0209 for 10 samples for forward and backward pass vs GMAE takes on average a mean time: 0.7044 seconds and a standard deviation: 0.0053 for 10 samples including forward, rendering, and backward pass. Which it added only a small overhard to the training, which recent advances in gslpat optimizations can also be incorporated to give faster training.
>
> **“Arranging the visualization of Gaussian layers from shallow to deep depths”**
>
> Thanks for this suggestion, we have added figures from shallow to deep in the appendix on lots of unfiltered samples.

---

> > ### Comment · Reviewer_PWCZ · 2024-11-24
> >
> > Thank you for your responses! However, I am not entirely clear on how the residual gaussian process works. Specifically, how does a k-layer MLP project changes from 256 gaussians to k*256 gaussians? Could you please elaborate more on this process? And during inference, how does this residual gaussian work? Will this introduce significant additional computation/latency?

---

> > > ### Author Response · Authors · 2024-11-24
> > >
> > > Thanks for your question. We have added a figure in the updated paper, in section A2 figure 13. The residual MLPs act on the latents from the decoder to project small changes to the initial Gaussians. The decoder produce (256 * d) vector (d is the hidden_dim of the decoder). In our pretrained model we only had one MLP-layer to project this to (256 * 14) Gaussians. Now, after this model is pretrained, we add $k$ new MLP-layers each with the same as before (to project from d to 14 dim). New Gaussians are taken as initial Gaussians + small changes learned by the residual heads. This process is also explained in section A2 figure 13.
> > >
> > > In Terms of computation, this does not add any significant overhead, since we only had few small mlp heads (in our case maximum 16), and the splatting and rendering is didnt affected by more Gaussians.  GMAE (with 256 Gaussians) takes on average a mean time: 0.7044 seconds and a standard deviation: 0.0053 for 10 samples including forward, rendering, and backward pass. GMAE (with 256*16=4096 Gaussians) takes on average a mean time: 0.7093 seconds and a standard deviation: 0.0054 for 10 samples including forward, rendering, and backward pass. This is not a significant increase in compute. However, we still need at least 1 epoch of finetuning, to get a better rFID than the pretrained model. But this can be sped up with better hyper-parameters.

---

### Official Review · Reviewer_uCko · 2024-11-03

**Soundness:** 3
**Presentation:** 3
**Contribution:** 1
**Rating:** 5
**Confidence:** 4

**Summary:**

The paper proposes to replace the decoder module of MAE with a gaussian splatting decoder≤µt: instead of decoding query pixel patches, it proposes to predict the parameters of 3D gaussian splats, which are then rendered onto a static camera position. The authors provide ablations on the number of gaussians used. They show several zero-shot capabilities which arise from such design: background/foreground segmentation and edge detection. The paper is well written and have high quality visualizations. They also analyze some properties of how the gaussians get distributed:

**Strengths:**

- I guess the main strength is some zero-shot capabilities, like foreground/background separation and edge detection.
- Despite to unconventional design, it does not lead to the loss of the main quality of self-supervised methods:
- The overall idea is quite unusual which, I believe, is a good quality of a scientific paper.
- Writing is very clear and the presentation quality is high.

**Weaknesses:**

- The method looks very unnatural and simply combines 2 popular ideas: 3d gaussians and MAEs. There are no particular advantages or insights in combining them. I feel the benefits are marginal and not worth the complications of the design.
- Zero-shot capabilities are not convincing: there are easier ways to obtain them with a higher quality (e.g., generative methods or generative multi-plane images with similar layered representations).
- The main advantage I would hope to see is having some 3D capabilities, but they are lost due to rendering from a static position.

**Questions:**

- What exactly is the main advantage of the proposed model? For which use-case would one realistically chose to use it?
- How much slower each training iteration has become compared to MAE?

---

> ### Author Response · Authors · 2024-11-22
>
> We thank the reviewer for comments on our well written paper, high quality visualizations, and good quality of a scientific paper. Here we answer the questions the reviewer is asking, and we are happy to answer more questions or run more experiments for the rebuttal.
>
> **“The method looks very unnatural and simply combines 2 popular ideas: 3d gaussians and MAEs.”**
>
> We agree with the reviewer, these are two bit orthogonal approaches. However, we'd like to cast the comment with positive light, being **very unnatural** meaning that we are doing something novel, and we showed that Gaussian and MAE can be integrated into the same framework. If **simple** combination can already show interesting results, it would be valuable to explore and share these findings with the community.
>
> our combination is unique and we showed some interesting results. We were interested in learning from large scale images, and MAE is proven to be a best approach to learn good **semantic** representations, on the other hand, Gaussians are good way to learn **structure/geometry**. A combination of these two approaches allowed us to learn good semantic representations (as shown in ImageNet, COCO experiments), as well as some geometry based on 2.1d representations (layering and edge detections).
>
> In addition to this, our new results on depth estimation shows it is more competitive than dinov2, and also our new results on scaling the number of gaussians shows this could be helpful with image generation tasks.
>
> **“Zero-shot capabilities are not convincing”**
>
> Our goal is to demonstrate that  these capabilities emerge purely from self supervised pre training objectives, without any task specific design choices. None of the results were explicitly trained for the zero-shot tasks. While we agree that task-specific models which were trained for these tasks might get better results, we only showed few zero-shot capabilities by only training on self-supervised objectives. We believed there might be more tasks which can be explored, from our models. We only showed a few examples, but since these are tasks the model was not optimized for, there might be more tasks that can be done with our models. Could the reviewer please clarify regarding "generative methods" or "generative multi-plane images", which shows zero-shot capabilities also emerge while the models are not specifically pre-trained for those tasks.
>
> **“I would hope to see is having some 3D capabilities”**
>
> it would be impossible to get full 3D representation from a large scale collection of 2D images from different scenes. We aim to get to some level higher than 2D and learn a 2.1D representation which would allow us to learn layering of the objects and scene. Hopefully, with addition of videos to the training data, we may be able to learn some slightly higher than 2.1D representations. We leave this exploration for future work.
>
>
> **“What exactly is the main advantage of the proposed model? For which use-case would one realistically choose to use it?”**
>
> We agree the tasks we have are not fully utilized, and if we care about a single zero shot task, we can simply use the best methods available there to get better performance. But rather, this work shows there is another way to train self-supervised models and how to get zero-shot capabilities from these models. From this we showed, a subset of zero-shot capabilities emerge from GMAE models, but this is not limited, and this opens more possibilities to explore more zero-shot capabilities in self-supervised models as a meta task.
>
> In addition to this, we also showed that in depth estimation case GMAE performs better than dinov2 and same as mae self-supervised models. We also showed a new way to increase the number of gaussians by 16x factor, by learning residual gaussians. This could be used as a better  initialization for gaussian splatting, or for generative models.
>
> Finally, we wanted to address that, equipped w/ Gaussians, we can unlock the use of decoders after MAE pre-training. Traditional SSL methods only care about the encoders; we show that with the Gaussians in the decoder, we can actually use the Gaussians to do more tasks (zero-shot). Of course these are some initial positive signals, but we believe these signals are interesting and worthy of sharing to the community.
>
>
> **“How much slower each training iteration has become compared to MAE?”**
>
> For MAE training on V100 gpus, takes on average time: 0.6471 seconds and Standard deviation: 0.0209 for 10 samples for forward and backward pass vs GMAE takes on average a mean time: 0.7044 seconds and a standard deviation: 0.0053 for 10 samples including forward, rendering, and backward pass. It added only a small overhead to the training, which recent advances in gslpat optimizations can also be incorporated to give faster training.

---

> > ### Comment · Reviewer_uCko · 2024-11-25
> >
> > I am thankful to the authors for their thorough response. Let me follow up on the discussion and clarify some of the raised concerns:
> >
> > > **The method looks very unnatural and simply combines 2 popular ideas: 3d gaussians and MAEs.**
> >
> > I highly value *unusual* (surprising) ideas, but when describing the "MAE + 3DGS" as unnatural, I did not see this in a positive light. I believe that an "A + B" type of a paper is influential only when the synergy brings a lot of benefits (e.g., MAE is "image transformer + masked pretraining" and showed unexpected simplicity and high quality, ViT is "transformer + image classification" and showed unexpected scalability, etc.). But with the current results, it's unclear why doing it. One can come combine any "A" (self-supervised learning, continual learning, meta-learning, few-shot learning, multi-task learning, curriculum learning, adversarially robust training, training in a latent space of a VAE) with any "B" (3DGS, NeRF, long videos, megapixel images, protein structures, table-based data structures, etc.) and do a research project with it. Whether this project would be insightful or not depends on the results. For the current submission, maybe I am just short-sighted, but I just do not see why exactly it's worth combining 3DGS and MAE and what this opens for the community.
> >
> > > **Zero-shot capabilities are not convincing**
> >
> > I have not found any substantially improved performance in the downstream applications which would urge the community to switch to the proposed setup from other designs. If one trains a diffusion model on ImageNet, they can do segmentation/depth estimation on top of its representations very accurately. For GMPI, I meant this work: Zhao et al., "Generative Multiplane Images: Making a 2D GAN 3D-Aware".
> >
> > > **I would hope to see it having some 3D capabilities**
> >
> > If the goal is to get a 2.1D representation, then why not just using a layered image representation (like an MPI) directly? I think that would give both edges and depth in the same manner.
> >
> > > **What exactly is the main advantage of the proposed model**
> >
> > In the response to this concern, the authors mentioned several potential downstream applications, but it's unclear whether these applications are actually achievable with the proposed design. The authors claim that GMAE can be useful for generative modelling, gaussian initialization, etc., but there is no evidence for that.
> >
> > Also, could the authors please provide a pointer for "We also showed a new way to increase the number of gaussians by 16x factor, by learning residual gaussians"? I have not found it in the latest manuscript version.
> >
> > > **How much slower each training iteration has become compared to MAE**
> >
> > This concern has been fully addressed and I don't have it anymore. I am thankful for the clarification.

---

> > > ### Author Response · Authors · 2024-12-04
> > >
> > > **gaussians by 16x**: We have added a figure in the updated paper, in section A2 figure 13. The residual MLPs act on the latents from the decoder to project small changes to the initial Gaussians. The decoder produce (256 * d) vector (d is the hidden_dim of the decoder). In our pretrained model we only had one MLP-layer to project this to (256 * 14) Gaussians. Now, after this model is pretrained, we add k new MLP-layers each with the same as before (to project from d to 14 dim). New Gaussians are taken as initial Gaussians + small changes learned by the residual heads. This process is also explained in section A2 figure 13.
> > >
> > > In Terms of computation, this does not add any significant overhead, since we only had a few small mlp heads (in our case maximum 16), and the splatting and rendering was not affected by more Gaussians. GMAE (with 256 Gaussians) takes on average a mean time: 0.7044 seconds and a standard deviation: 0.0053 for 10 samples including forward, rendering, and backward pass. GMAE (with 256*16=4096 Gaussians) takes on average a mean time: 0.7093 seconds and a standard deviation: 0.0054 for 10 samples including forward, rendering, and backward pass. This is not a significant increase in compute. However, we still need at least 1 epoch of finetuning, to get a better rFID than the pretrained model. But this can be sped up with better hyper-parameters.
> > >
> > > **why not just using a layered image representation**: We agree with the reviewer that MPI is another valid representation. In the same spirit Gaussians are also another valid representation. The scope of this work is not to compare these intermediate representations but rather show the benefits of using Gaussians as intermediate representations.
> > >
> > > **main advantage**: We agree with the reviewer that we don't have evidence of very strong applications yet, but we hope this would need more exploration and our work is useful for the research community as a first step.

---

> > > > ### Author Response · Authors · 2024-12-04
> > > >
> > > > **"MAE + 3DGS"**:
> > > >
> > > > Thank you for sharing your detailed perspective. We appreciate the value of impactful A + B papers and agree that their influence lies in demonstrating meaningful synergies between the components. This principle has guided our efforts in this work, and we would like to provide further clarity on why we believe the combination of MAE and 3DGS presents a valuable and insightful contribution.
> > > >
> > > > MAE excels as a powerful framework for representation learning, particularly for the encoder. However, its design inherently discards the decoder post-pretraining, leaving its potential for downstream tasks unexplored. In contrast, 3DGS is a lightweight yet effective framework for 3D reconstruction, with untapped potential in representation learning. By integrating 3DGS into the decoder of MAE, our proposed GMAE synergistically combines the strengths of both approaches, resulting in:
> > > > - **Zero-shot capabilities** within the decoder, a novel property that the standard MAE framework does not enable.
> > > > - **Enhanced representation learning for 3DGS**, unlocking its utility beyond traditional 3D reconstruction tasks.
> > > >
> > > > We believe these results reflect a meaningful synergy, showcasing how the strengths of MAE and 3DGS complement each other to address their respective limitations.
> > > >
> > > > Regarding your comment on the “unnatural” combination, we apologize for the misinterpretation and casted our excitement into the response. On the other hand, while we agree the insightfulness depends on the results, we also want to respectfully point out that the interpretation of the results can vary among researchers. From our perspective, GMAE not only provides novel capabilities but also addresses gaps in both constituent methods, which we believe advances the field.
> > > >
> > > > Finally, while we acknowledge that many potential A + B combinations are possible, such possibilities *do not diminish* the contributions of our work. Each combination must be evaluated on its own merits, and we hope the insights and illustrations provided in the work can already justify the *value* of GMAE for the research community.
> > > >
> > > > Thanks again for the feedback and holding us to a high standard.

---

### Official Review · Reviewer_qLSS · 2024-11-04

**Soundness:** 3
**Presentation:** 4
**Contribution:** 2
**Rating:** 6
**Confidence:** 3

**Summary:**

The paper introduces Gaussian Masked Autoencoders (GMAE), an extension of MAEs that integrates a learned intermediate Gaussian representation, rendered into an image using Gaussian splatting. This intermediate representation is shown to offer several benefits over fixed patches, such as foreground-background separation, edge detection, and image layering, while achieving performance on par with or even surpassing standard MAEs in image recognition and downstream tasks. The authors support their claims with a comprehensive suite of experiments.

**Strengths:**

- The paper explores an interesting topic of adding additional inductive biases to self-supervised image representation learning techniques.
- The writing is clear and well-structured.
- The experiments section includes a wide variety of downstream applications and comparisons.

**Weaknesses:**

- As talked about in the Discussion section, the number of Gaussians used in GMAE is significantly lower than the quantities typically used in scene reconstruction applications, where Gaussian splatting is well-known. This is because each Gaussian corresponds to a unique token in the lightweight decoder, so increasing their number would cause considerable slowdowns.
- Minor typo on Line 503: Fig 12 → Fig 11

**Questions:**

- The GMAE model assumes a fixed camera projection for its rendering. Therefore, it most likely does not need all 14 degrees of freedom that is normally used in scene reconstruction applications. Do you think there is any benefits to manually reducing these redundant DoFs when training such models?
- Since the gaussian representation introduces a 3D inductive bias for the model, it would be interesting to compare this model with MAEs in a depth-prediction downstream task. In theory, GMAE should be much better equipped for solving such tasks. Do you expect to see a significant improvement over MAEs, or would the limited number of gaussians not allow for such a thing?
- As discussed in Section 4.1, the decoder is decoupled from encoder tokens, allowing the number of Gaussians to be increased arbitrarily after training. However, it’s mentioned that four separate models were trained to decode 64, 128, 256, and 512 gaussians, respectively. How well does a decoder generalize to Gaussian counts other than the one it was trained on? For example, does a decoder trained with 256 gaussians achieve better performance is evaluated with 1024 gaussians?

---

> ### Author Response · Authors · 2024-11-22
>
> We thank the reviewer for their comments on our interesting approach, wide variety of experiments and clear writing. Here we answer the questions the reviewer is asking, and we are happy to answer more questions or run more experiments for the rebuttal.
>
> **“The number of Gaussians used in GMAE is significantly lower than reconstruction methods”**
>
> We agree with the reviewer, that the limit on  the number of gaussians is a limitation in the current model (it's a limitation as this is the first time someone combines Gaussians to a representation learning based framework), and the current design we are limited by memory. We want to point out that we are not focused on reconstruction quality here and the focus is on the meaning of the learned Gaussians. However, here we modify the architecture to show the potential of further increasing the number of Gaussians.
>
> We tried to solve this problem by learning more Gaussians step by step, as residual gaussians. First, we pretrained the model with 256 Gaussians, and after that, we initialize an k MLP layers, to project changes in 14 features to k*256 Gaussians. These MLP layers essentially learn the small changes from the main 256 gaussians. At the start these MLP layers are initialized with zero weights, hence they don't affect the reconstruction for the original 256 gaussians.
>
> With this setting, we were able to train up to 4096 gaussians, but not limited (we will update here with more gaussians), since it is not limited by the memory size anymore. Below we show the reconstruction FID as we increase the number of gaussians step by step.  Finally, we also fine tune the model (4096*) without masking and on full reconstruction, and it achieves 18 rFID, without perceptual loss or vae.
>
> | Number of Gaussians   | rFID   |
> |---------|--------|
> | 256         |  89.45  |
> | 1024       |  80.32  |
> | 4096       |  63.87  |
> | 4096*      |  18.71  |
>
>
>
>
> **“Any benefits to manually reducing redundant DoFs when training such models?”**
>
> The 14 DoFs includes 3 for location, 3 for color, 4 for rotation as quaternion, 3 for scale and 1 for opacity. If we treat all GSs as 2D instead of 3D living on the image plane, then we could reduce the DoFs to 11 (2 for location, 2 for rotation and 2 for scale).  But in this case we lose the ability to model the image as "layers" because all GSs effectively placed at the same depth level. Since, our aim is to have this extra degree of freedom to allow the model to place gaussians at different depth levels, so that we can get these emergent zero-shot capabilities. We will train a model with 2DGS, and get back before the discussion period ends.
>
> **“it would be interesting to compare this model with MAEs in a depth-prediction”**
>
> We thank the reviewer for this valuable feedback, and we have evaluated our models on depth estimation tasks. Please find the depth estimation results below, on NYU depth estimation tasks, GMAE outperforms dinov2 models and perform the same as MAE or slightly better.
>
> | Model   | RMSE   |
> |---------|--------|
> | DINOv2  | 0.4761 |
> | MAE       | 0.4345  |
> | GMAE    | 0.4336 |
>
>
> **“How well does a decoder generalize to Gaussian counts other than the one it was trained on?”**
>
> At test time we can only infer with a fixed number of gaussians with the current design, but we plan to explore training the models on Matryoshka style loss, which can allow us to use variable number of gaussians at test time and gives coarse to fine reconstructions as we increase the number of gaussians. We will add a discussion regarding test time scaling of the number of gaussians in the paper.

---

### Official Review · Reviewer_5Et3 · 2024-11-05

**Soundness:** 2
**Presentation:** 3
**Contribution:** 2
**Rating:** 3
**Confidence:** 4

**Summary:**

This paper proposes to use 3D Gaussians a-la Gaussian Splatting as a mid-level representation that are predicted from masked input patches a-la Masked Autoencoders. Training is done by splatting gaussians differentiably to render RGB from a fixed camera using MAE training losses (unnormalized). Authors pre-train a ViT-B for 400 epochs using this method and show a slight performance drop vs. regular MAE training on Imagenet classification and COCO instance segmentation.

They also show that the predicted gaussians are layered in depth from the fixed camera by frequency. The authors attempt to utilize this property to show some results on zero-shot edge detection and figure-ground segmentation, where they perform similarly or worse to a very selected set of baseline methods.

**Strengths:**

Originality:
The proposed method of using 3D gaussians as their intermediate representation is original and interesting. However, the related work section misses very related work and focuses on some more irrelevant topics (discussed more later)

Quality:
The proposed representation seems to learn better reconstructions compared to MAE. However, beyond this, I personally do not agree with the proposed evaluations to show the benefits of this representation. No comparisons are made to any other similar intermediate representations that could be thought of (discussed more later).

Clarity:
The paper is presented clearly and laid out well. Some statements are misleading (discussed more later).

Significance:
I believe this paper has potential to be significant to the community. However, the current results show that GMAE does not improve over MAE meaningfully (and is evaluated very sparsely across the design space) and the learnt gaussians are not very useful in the zero-shot case even in the tasks that the authors decide to focus on i.e. figure-ground segmentation and edge detection.

**Weaknesses:**

\textbf{Related Work:}
The paper does not talk about any related work on using mid-level representations in vision beyond using learned "tokens". The authors misrepresent MAE as only training for pixel reconstruction. MAE has an ablation experiment where they also use tokens to explore the "best of both worlds" approach that the authors suggest they take. MAE-VQGAN proposed in Bar et al. 2022 is also a tokenized MAE learner. Other mid-level representations can be thought of that are similar to this method. For example, one could directly predict a multi-plane representation and render it. One could use superpixels a-la superpixel sampling networks (Jampani et al.) as the mid-level representation. There is no discussion on other possible methods and prior mid-level representations used in vision. Other papers have proposed losses that learn self-supervised grouping, (which is one of the benefits according to the authors), such as those based on Slot Attention or Leopart (Ziegler et al, CVPR 2022). In the discussion, the paper claims -- "Nonetheless, we have shown that one no longer has to choose between pixels and latent representations for visual modeling.". This is misleading compared to related work as mentioned above.

\textbf{Why is the gaussian representation better?}
There are claims across the paper that the gaussian representation is better due to its efficiency (the proposed model is slower than MAE while performing worse), due to its non-isotropic representation vs. grids (no comparisons are made to back the claim that this is useful for pre-training). The only real benefit shown in the paper is that GMAE reconstructions are higher-fidelity as opposed to MAE. However, the authors immediately claim "L362: As a result, our reconstructions can be used directly for other tasks without needing to add a GAN loss or an upsampling layer on top." which is again unsubtantiated in the paper. Which other methods need a GAN loss or upsampling layer on top? The other tasks proposed here are figure ground segmentation and edge detection, where the model performs poorly overall. Discussed more in the next section.

\textbf{Frequency clustering in depth}
The authors make the following claims:

"This may be due to the fact that with random initialization, the points closer to the camera represent low-frequency information, while the points far from the camera model the high-frequency information"

"The layer-wise rendering highlights the model’s ability to separate objects and represent them in distinct frequency layers"

"In the real world, backgrounds tend to have low-frequency regions while objects usually have high-frequency details. This correlation leads to our zero-shot results."

These are incompatible claims and I think these are mis-leading when looking at the results. Objects are clearly not separated across frequencies. Low frequency shapes of most objects seem to be captured in the initial layers and higher frequencies of their shapes in later layers. Figure 6 and 7 corroborate this. Claiming that objects are separated and represented in distinct frequency layers does not appear true from the results and does not follow the prior claim of frequency based clustering. Individual instances of objects are not separated in any way. The edge detection results show lots of spurious edges coming from the gaussian representation which only make edge prediction worse. The argument that backgrounds tend to have low-frequency regions while objects.. is barely enough to make the claim that objects are separated in the model. The examples shown are few and relatively simple with one bird on a tree and clear background. Yet, the model is unable to separate the tree branches from the bird, and even the bird is not clearly segmented. I believe the assertion that frequency based depth ordering happens. The follow-up claim that this leads to emergence of objects or even parts is a stretch.


\textbf{Experimental details and strength of results}
The authors only train ViT-B for 400 epochs. The authors could have pre-trained for 1600 epochs, or tried a ViT-L architecture. Currently there is no clarity whether this approach will scale to a larger ViT or if it will continue to improve with additional training as MAE does. The ablation studies over c, masking ratio and loss masking, normalization and the usage of batch size 4096 show that sufficient GPU resources were used in pre-training. At least pre-training ViT-B till 1600 epochs should have been possible for the authors. It would be very useful to add these results. Without these results, it is impossible to verify whether GMAE scales as MAE does.

For the figure-ground segmentation results, there are no details on the experiment. What layer was used for figure ground segmentation in the layering? No discussion on the baselines is presented. Models such as Leopart (Ziegler et al.) need to be compared. Their results on zero shot segmentation are way more advanced while not needing a sparse gaussian representation that the authors claim is the reason why their figure ground segmentation results are strong.

The edge prediction results are worse than using a Sobel filter for edges. There are clearly numerous spurious edges in the qualitative result that probably come from gaussians that represented interior regions of objects that do not correlate with any real edges.

**Questions:**

Please address the issues brought up in the weaknesses section.

Experimental results on ViT-B trained to 1600 epochs and ViT-L could be very useful, but I understand that it is unreasonable to ask for these results within the rebuttal period.

I believe the authors need to focus more carefully on their evaluation. If depth based frequency layering is all that the model achieves over standard MAE, could this be done without using this intermediate representation? What other tasks can be helped by such a representation? Clearly figure-ground segmentation and zero-shot edge detection do not benefit from this method.

---

> ### Author Response · Authors · 2024-11-22
>
> We thank the reviewer for comments on the ”paper is original and interesting”, “paper has potential to be significant to the community” and “paper is presented clearly and laid out well”. Here we answer the questions the reviewer is asking, and we are happy to answer more questions or run more experiments for the rebuttal.
>
> **Related works**
>
> We agree that our proposed mid-level representation is but one of many existing options, with many more to come. While we cited several of these related works, we will also make sure to modify the introduction and related works to further highlight this fact in the updated version of the submission.
>
> We agree with the reviewer that there are various options for mid-level representations. We thank the reviewer for pointing out Superpixel Sampling Networks (SSN), and we will discuss them in the related works section. However, there are a few differences we would like to point out. While there is a similarity between gaussians and superpixels as intermediate representations, unlike SSN, our models are trained for self-supervision without any task specific data. Edges and segments are extracted from our model, rather than trained for end-to-end as in Superpixel Sampling Networks.
>
> Our approach differs from SSN or Slot attention in several ways. Unlike SSN or Slot attention, we don't utilize any specific loss function, and we simply use L2 reconstruction loss. In terms of architecture, we don't add any specific changes to the model. In addition to this we also show that our method can learn representations that are both good for recognition and detection as well as mid-level zero-shot problems.
>
> We also cited Bar et al. in our main paper, discussed it in the related works section, and compared to it in table 3. We found that our method outperformed Bar et al. in figure ground tasks. In addition to that, bar et al only produce discrete tokens, and it is hard to reason about or manipulate the discrete token, since they don't have any explicit representation.
>
> "Nonetheless, we have shown that one no longer has to choose between pixels and latent representations for visual modeling." — based on the additional related works discussed above, we see this can be misleading and we will remove this in the paper.
>
>
> **“Why is the gaussian representation better?”**
>
> We believe that, GMAE adds extra capabilities while persevering the representation quality along the MAE axis. We explored new directions such as layering, segmentation, and edge detection. The rebuttal also includes new experimental results on metric depth estimation where GMAE performs better than dinov2 on NYU depth estimation. We also explored the generation quality of GMAE and with new changes to the architectures, we were able to train with 4096 gaussians and more, without memory issues. All these additional capabilities are due to the use of gaussians as an intermediate representation, and it that could open up new capabilities in the future. Below we address a few concerns regarding the GMAE model.
>
> **Slower**: GMAE is not significantly slower than MAE. For MAE training on V100 gpus takes on average: 0.6471 seconds with standard deviation: 0.0209 for 10 samples for forward and backward pass. GMAE takes on average: 0.6944 seconds and a standard deviation: 0.0053 for 10 samples including forward, rendering, and backward pass. This adds only a small overhead to training. Recent advances in GSplat optimizations can also be incorporated to result in faster training.
>
> **GAN loss**: The MAE github mentions that an addition of a GAN loss leads to better reconstructions. According to off-line communications with MAE authors, the additional adversarial training only improves reconstruction quality (the images become sharper), but it does not improve representation quality. Bar et al, uses discrete tokens from VQ-GAN and is trained with a GAN loss to have sharp reconstructions.
>
> **Frequency clustering in depth**: “Individual instances of objects are not separated in any way” — this is true, and we do not claim this, as based on the training recipe it is **unlikely** that the model would be able to group based on instances. Our claim was “separate objects in the z direction”, this is layering rather than instance segmentations. We agree with the reviewer that these are not explicit segmentation rather, reasonable reconstructions of different layers.

---

> > ### Author Response · Authors · 2024-11-22
> > **Official Comment by Authors --- continued**
> >
> > **Experimental details and strength of results**
> >
> > **The authors only train ViT-B for 400 epochs.**: We agree that to verify the scalability of this approach, we should train bigger models for longer, and we will start running this experiment and should be able to get the results by the end of this discussion period.
> >
> > **figure-ground segmentation results**: We thank the reviewer for pointing Leopart (Ziegler et al.), which is very relevant. We will try to run our models on these baselines, and get results before the end of the discussion period. We also want to add that, the training objectives of Leopart are designed for clustering, and with self-supervised training they achieve great results. However, we would like to point out that our objective is not specifically designed for clustering.
> >
> > **zero-shot edge detection does not benefit from this method.** --- Our claim in this paper is not that we improve upon SOTA on these tasks. Rather, our goal is to demonstrate that by just employing a self-supervised MAE loss, we can get these capabilities to emerge without designing specific datasets or architecture for each of these tasks individually. For edge detection, the model is not trained for this task, and the objective does not enforce edge detection.

---

> > > ### Comment · Reviewer_5Et3 · 2024-11-26
> > >
> > > Thanks to the authors for their rebuttal.
> > >
> > > Re: mid-level representations, I should have also mentioned multi-plane images (see Tucker et al. as an example). The novel-view synthesis literature has explored various mid-level representations over the years that can be re-rendered from novel views. Gaussian Splatting and Nerfs also descendants from work in this literature. A multi-plane image representation might have all the same depth layering benefits that are proposed here.
> > >
> > > Arguing that GMAE doesn't use "specific losses" like SSNs or Slot Attention definitely says that the proposed method is simpler in comparison. However, it has nothing to say about whether it is better since all these methods are self-supervised at the end of the day. As an analogy, DinoV2 is widely used now and is far from simple in terms of the different losses used. However, they show how each component of their loss is important and its impact in the community is undeniable. So I hope the authors understand why I do not buy into this distinction.
> > >
> > > My argument about frequency clustering is still valid. In the layers, it can be seen that it's not just depth, but the different frequency parts of objects that are differentiated in depth. In Figure 7 (which are all very simple examples of a single bird on a clear background), it can be seen that low-frequency colours on the birds show up in early layers, followed by high frequency details in later layers. Given how gaussian splats are rendered and because GMAE does not need to model multi-view accuracy, it makes sense that the gaussians are not placed on direct geometry but instead lower frequencies are placed closer to the camera followed by higher frequencies away from the camera. This is also clear since the sky and background shows up in earlier layers.
> > >
> > > Thanks for adding the depth estimation results! It's great to see that GMAE beats DinoV2 there. However, the fact that there is no significant difference from MAE is also concerning. The major argument across the paper is that there is emergent depth layering, shown through experiments where the results are far from convincing. However, if there is no direct effect on improving downstream depth estimation itself, then the argument in the paper becomes much weaker.
> > >
> > > Overall, the results in this paper are far from convincing that there is any benefit to using single image Gaussian predictions that are splatted for rendering for representation learning. For edge detection, the authors say, "Rather, our goal is to demonstrate that by just employing a self-supervised MAE loss, we can get these capabilities to emerge without designing specific datasets or architecture for each of these tasks individually". I would like to clarify here that the authors have not shown any useful edge detection capabilities that cannot be solved with a simple filter. These filters are not SOTA, they are extremely simple image processing methods.
> > >
> > > Overall, I would like to reiterate that the idea in this paper is novel. However, I haven't seen experimental evidence in this paper that this novel idea is useful. I believe it needs more work and exploration to find merits of this work, which are not in figure-ground segmentation or edge detection in the way that is presented in the paper. Depth estimation is a great addition and the early results show no significant difference to MAE.
> > >
> > > I haven't seen any evidence here to improve my proposed score of 3 and urge the authors to work further on this paper, since it is an interesting idea that perhaps needs a different application than proposed here.

---

> > > > ### Author Response · Authors · 2024-12-04
> > > >
> > > > **Related works and Introduction:** We have fully revised the paper to accommodate the changes from the reviewer. Please see the updated manuscript. We have included super pixel sampling networks, multi-plane images, leopart clustering and slot-attention.
> > > >
> > > > *Mid-level Representations: Image can be constructed by operating functions on some representations. One line of approach keeps the representations in the latent spaces, and uses a pretrained decoder network to reconstruct the image. VAE (Kingma, 2013) with image synthesis (Rombach et al., 2022; Li et al., 2023) are good examples of this case, along with MAE He et al. (2022) and BEiT Bao et al. (2021). Other lines of approaches follow structured representations to represent an image. There are various such options: super-pixels, Gaussians, SVG code, and multi-plane images, etc. For example, Super-pixel Sampling Networks (Jampani et al., 2018) learns to predict super-pixels as the representation to reconstruct and to predict segmentations and flow. Multi-plane images is another way to represent an image (Tucker & Snavely, 2020), where an image is composed of multiple layered planes and can be learned end-to-end. There are hybrid approaches also. For example, Slot Attention (Locatello et al., 2020) learns an intermediate representation for objects by adding a bottleneck in the model architecture. Similarly, Leopart (Ziegler & Asano, 2022) learns to cluster the patches based on self-supervised clustering. In this paper, we take another approach which uses 3D Gaussians as intermediate representations to reconstruct an image.*
> > > >
> > > > **1600 epoch results**: We have pretrained a vit-l model on 1600 epochs, and then fine tuned for imagenet classification. This model achieved 85.0% on imagenet classification accuracy. This shows that our model does scale with model size and training epochs
> > > >
> > > > **specific losses**: We agree with the reviewer that utility outweighs here on Dino case. We believe that our approach is the first step towards learning both semantics and geometric representation and we hope this will open up further research opportunities.
> > > >
> > > > **Frequency based clustering**: We also agreed on this on the paper. We showed that the correlation on depth and frequencies are the reason we get a layering effect. This is explained in section 4.4.
> > > >
> > > > **MAE vs GMAE on depth**: As mentioned in the paper, our goal is to not get better than MAE, but rather get more capabilities of vision models, via self-supervised pre-training.

---

### Meta-Review · Area_Chair_6S5U · 2024-12-17

**Metareview:**

The paper introduces a novel combination of MAE and Gaussian Splatting but fails to demonstrate significant advantages over standard MAE in key benchmarks, with zero-shot results like edge detection and figure-ground segmentation remaining weaker than simple baselines. The proposed Gaussian representation, while intriguing, the claimed frequency-based depth layering remains unconvincing. Additionally, the evaluation on outdated datasets and limited comparisons with state-of-the-art methods further undermines the practical impact and scalability of the approach.

**Additional Comments On Reviewer Discussion:**

During the rebuttal, reviewers raised concerns about limited comparisons to state-of-the-art methods (e.g., DINOv2, SAM) and outdated datasets, weak zero-shot task performance, and unclear justification for using Gaussian Splatting over alternatives like NeRF or MPI. The authors addressed these by adding depth estimation results (outperforming DINOv2), scaling Gaussian representations to 4096 Gaussians with residual MLPs, clarifying the emergent layering effect, and committing to further evaluations on modern datasets. While these updates demonstrated scalability and emergent properties, I agree with the reviewers and I am  unconvinced about the practical advantages and broader impact of the approach.

---

### Decision · Program_Chairs · 2025-01-22

Reject